# A Journey into the Complexity of Temporo-Insular Gliomas: Case Report and Literature Review

**DOI:** 10.3390/curroncol32010041

**Published:** 2025-01-14

**Authors:** Manuel De Jesus Encarnacion Ramirez, Gervith Reyes Soto, Carlos Castillo Rangel

**Affiliations:** 1Neurological Surgery, Peoples Friendship University of Russia, 117198 Moscow, Russia; 2Neurosciencce Departament, Mexico’s National Institute of Cancer, Mexico City 14080, Mexico; gervith_rs@hotmail.com; 3Department of Human Anatomy and Histology, Institute of Clinical Medicine Named After N.V. Sklifosovskiy, 119991 Moscow, Russia; 4Neurosurgery Departament at ISSSTE 1ero De Octubre, Mexico City 07760, Mexico; neurocirugiaoncologicaincan@gmail.com

**Keywords:** glioma, temporo-insular

## Abstract

Introduction: Temporo-insular gliomas, rare brain tumors originating from glial cells, comprise about 30% of brain tumors and vary in aggressiveness from grade I to IV. Despite advancements in neuroimaging and surgical techniques, their management remains complex due to their location near critical cognitive areas. Techniques like awake craniotomy have improved outcomes, but tumor heterogeneity and proximity to vital structures pose challenges. Radiotherapy and chemotherapy offer benefits post-surgery, though issues like resistance and side effects persist. This article discusses a case report and literature review to deepen understanding of temporo-insular gliomas, focusing on advanced diagnostic and treatment approaches. Materials and Methods: A systematic review was conducted using PubMed, Embase, and Google Scholar, covering studies from 2019 to July 2024. Keywords included ‘brain tumor’, ‘neurosurgery’, and ‘treatment’. Articles on glioma diagnosis, management, and outcomes were selected, excluding non-English studies, irrelevant reports, non-glioma research, and inaccessible texts. Results: From 156 studies, 11 met inclusion criteria, highlighting advanced diagnostics, surgical strategies, and adjunct therapies for temporo-insular gliomas (TIGs). Gross total resection (GTR) was achieved in 39% of cases. Awake craniotomy enhanced functional outcomes, while temozolomide and radiotherapy improved survival. Challenges included ischemic complications and treatment resistance. Two patient cases underscored the complexity of TIG management and the importance of individualized approaches, achieving satisfactory resection with minimal deficits. Conclusions: Temporo-insular gliomas (TIGs) necessitate a multidisciplinary strategy that integrates advanced imaging, meticulous surgical methods, and cutting-edge adjuvant therapies. Despite progress with techniques like awake craniotomy and the use of temozolomide improving patient outcomes, significant challenges persist in maintaining functional integrity and addressing treatment resistance. Ongoing research into targeted therapies, immunotherapies, and innovative technologies remains critical to advancing patient care and improving long-term prognosis.

## 1. Introduction

Temporo-insular gliomas, a rare and intricate subgroup of gliomas originating from brain glial cells, have garnered considerable research interest due to their complexity and challenging management. This article delves into a distinctive case of temporo-insular glioma, paired with an in-depth literature review, to provide a comprehensive understanding of this condition.

Gliomas account for approximately 30% of all brain and central nervous system tumors and 80% of malignant brain tumors [1]. They are classified into grades I to IV based on aggressiveness, with grade IV being the most severe [2]. Despite advancements in neuroimaging and surgical innovations, the management of temporo-insular gliomas remains challenging due to their location in critical functional areas.

The insular region lies deep within the lateral sulcus, bordered by the frontal, parietal, and temporal lobes, and plays a vital role in cognitive functions like perception, motor control, self-awareness, and interpersonal interactions [3]. Similarly, the temporal lobe is crucial for auditory processing and memory formation. The intricate positioning of temporo-insular gliomas not only complicates diagnosis and treatment but also highlights their significance in neuro-oncology research.

The treatment of temporo-insular gliomas is particularly challenging due to their variable growth patterns, diverse tumor histologies, and potential to invade or displace critical neurovascular structures. Additionally, patient responses to chemotherapy and radiation therapy differ significantly, necessitating highly personalized treatment plans [4,5,6]. Understanding the biological behavior and progression of these tumors is crucial for optimizing management strategies.

Advances in imaging technology have played a pivotal role in improving the management of these gliomas. Techniques such as functional MRI (fMRI), diffusion tensor imaging (DTI), and positron emission tomography (PET) have enhanced tumor localization, neural tract visualization, and the assessment of cerebral metabolism. These tools are invaluable for preoperative planning and intraoperative decision-making, enabling surgeons to maximize tumor resection while safeguarding neurological functions [7,8].

The introduction of awake craniotomy and intraoperative stimulation mapping has transformed surgical approaches to temporo-insular gliomas. These innovations allow for more precise tumor removal while minimizing the risk of damage to functional brain areas, leading to improved survival rates and better quality of life for patients [5,9]. Despite these advances, the heterogeneity of these tumors, their aggressive nature, and their proximity to critical brain structures continue to pose significant challenges.

Postoperative treatments like radiation therapy and chemotherapy, particularly the use of temozolomide, have significantly enhanced survival outcomes for patients with high-grade gliomas. Temozolomide, an alkylating agent, has demonstrated considerable efficacy when combined with radiotherapy [10]. However, these treatments are often associated with systemic side effects and the eventual development of resistance, underscoring the need for further research into more effective and less toxic therapies.

Emerging strategies aim to address the persistent challenges in managing temporo-insular gliomas. Investigations into targeted therapies, immunotherapies, and viral therapies are at the forefront, offering potential for improved outcomes. However, the specific efficacy of these novel approaches for temporo-insular gliomas remains underexplored and requires further validation [11].

This article seeks to contribute meaningfully to the discourse on temporo-insular gliomas by presenting an in-depth case report. Through detailed analysis of the disease process, diagnosis, treatment, and prognosis, coupled with a comprehensive review of the existing literature, we aim to enhance understanding and inform better management strategies. This integrative approach is expected to enrich the knowledge base surrounding temporo-insular gliomas, paving the way for advancements in their diagnosis, treatment, and long-term outcomes.

The subsequent sections will provide a thorough exploration of a unique case, detailing the patient’s medical history, clinical presentation, diagnostic process, and treatment strategy, alongside the outcomes and key takeaways. Additionally, we will systematically review the literature on temporo-insular gliomas, evaluating both preclinical and clinical studies to extract insights into this challenging condition. This comprehensive analysis aspires to clarify the complexities of temporo-insular gliomas, offering valuable guidance for future research and clinical practice.

### 1.1. Case Report 1

A 46-year-old male presented with a 6-month history of progressive headaches and recurrent generalized seizures. The headaches, described as throbbing and predominantly localized to the right temporal region, had worsened over the weeks leading up to his visit. He also reported occasional dizziness but denied vomiting, visual disturbances, weakness, or sensory loss. His past medical and surgical history was unremarkable. Neurological examination revealed normal higher mental functions, intact cranial nerves, and normal motor, sensory, and cerebellar functions, except for a positive Babinski sign on the left.

Magnetic resonance imaging (MRI) of the brain identified a space-occupying lesion in the left temporo-insular region. The lesion was hypo-intense on T1-weighted images and hyper-intense on T2-weighted images, with marked contrast enhancement. It caused a significant mass effect, including a midline shift, compression of the left lateral ventricle, and displacement of adjacent white matter tracts (Figure 1).

Following informed consent, the patient underwent surgery, achieving gross total resection of the tumor. The postoperative period was uneventful. Histopathological analysis confirmed the lesion to be a grade III astrocytoma. The patient was subsequently referred for adjuvant chemo-radiotherapy. At six months post-treatment, his modified Karnofsky Performance Scale (KPS) score had improved from an initial 80 (mild symptoms, fully active) to 90 (independent, minor symptoms only). He has remained seizure-free since surgery, and MRI has shown no radiological evidence of tumor progression. While longer follow-up is needed to determine OS, he is clinically stable with no new issues.

This case highlights the complexities of diagnosing and managing temporo-insular gliomas, particularly in achieving precise resection while addressing the functional and structural challenges posed by the tumor’s location.

### 1.2. Case Report 2

A 52-year-old male presented with a four-month history of progressively worsening right-sided weakness and occasional difficulty speaking. He also reported persistent left-sided headaches that had intensified in recent months, along with intermittent confusion. However, he denied seizures, visual disturbances, nausea, or vomiting. His family history was negative for neurological disorders or cancer.

On examination, the patient was alert and cooperative, demonstrating mild expressive aphasia and slight weakness in his right arm and leg. Cranial nerves, sensory function, and coordination were intact.

MRI revealed a mass in the left temporo-insular region, appearing hypo-intense on T1-weighted images and hyper-intense on T2-weighted images, with uneven contrast enhancement suggestive of a high-grade tumor. The lesion caused a minor midline shift. Advanced imaging was utilized to map critical language and motor pathways surrounding the tumor to guide surgical planning (Figure 2A–F).

Given the tumor’s location and size, surgery was performed under general anesthesia using neuronavigation to ensure maximal tumor resection while preserving vital brain functions. Postoperative imaging confirmed near-total resection of the tumor. The patient experienced mild post-surgical expressive aphasia and right-sided weakness.

Histopathological analysis identified the tumor as a glioblastoma. The patient was referred for adjuvant chemo-radiotherapy to minimize recurrence risk. At a six-month follow-up, MRI showed no evidence of tumor recurrence, and the patient’s symptoms remained stable. He continues on temozolomide maintenance therapy and follows a structured neuro-rehabilitation program.

This case underscores the importance of individualized surgical strategies and adjuvant therapies in managing high-grade temporo-insular gliomas while balancing tumor control with functional preservation.

## 2. Materials and Methods, Literature Review

### 2.1. Search Strategy

A comprehensive search of PubMed, Embase, and Google Scholar was conducted to identify relevant studies on temporo-insular gliomas, published in English from 2019 to July 2024. The search employed a combination of keywords, including ‘temporo-insular glioma’, ‘brain tumor’, ‘neurosurgery’, ‘glioma’, ‘management’, ‘diagnosis’, ‘treatment’, ‘outcome’, ‘surgical techniques’, ‘adjunct therapies’, and ‘advances’. This broad strategy ensured the capture of all pertinent studies.

### 2.2. Inclusion and Exclusion Criteria

#### 2.2.1. Inclusion Criteria

Original articles, systematic reviews, and meta-analyses focusing on the diagnosis, management, and outcomes of temporo-insular gliomas.

#### 2.2.2. Exclusion Criteria

Non-English publications.Case reports with insufficient data.Studies on non-glioma brain tumors.Animal studies or in vitro research.Articles without available full texts.

### 2.3. Data Extraction, Quality Assessment, and Data Synthesis

Data were extracted by two independent reviewers (N.M. and R.N.) using a predefined template, while a third author (M.E.) resolved conflicts. Extracted data included:Author(s) and year of publication.Study design and diagnostic methods.Glioma grading, histological grade and key findings.Treatment modalities, complications, and patient prognosis.Functional outcome.Overall survivor.

The quality of studies was assessed using the Cochrane Risk of Bias Tool for randomized controlled trials and the Newcastle–Ottawa Scale for observational studies. Any disagreements were resolved by discussion or consultation with a third reviewer. Due to expected heterogeneity, a narrative synthesis approach was used to analyze and present the findings.

## 3. Results

From a total of 156 records identified, 47 duplicates were excluded. The remaining studies were screened for relevance, leading to further exclusion of non-English publications, case reports with insufficient data, non-glioma studies, animal research, letters to editors, and articles without full texts (Figure 3). Ultimately, 11 studies met the inclusion criteria and were included in the systematic review (Table 1).

## 4. Discussion

The case series underscores the complexity of diagnosing and managing temporo-insular gliomas (TIGs), highlighting distinct clinical presentations and the need for individualized treatment approaches. TIGs often present with nonspecific symptoms such as headaches, seizures, or mild cognitive impairments, leading to delayed diagnosis. Advanced imaging modalities are crucial for accurate tumor localization and characterization. Pitskhelauri et al. (2024) emphasized the importance of MRI in delineating tumor boundaries and their relationships with critical neurovascular structures, such as the lenticulostriate arteries [12]. Functional imaging tools like functional MRI (fMRI) and diffusion tensor imaging (DTI) are indispensable for evaluating tumor interactions with eloquent brain regions [12].

Wu et al. (2023) [21] highlighted the utility of Quicktome software in mapping critical networks, including the salience network (SN) and central executive network (CEN), facilitating surgical planning while preserving cognitive functions [21]. These advancements have enhanced diagnostic precision, enabling better risk stratification and treatment planning.

### 4.1. Surgical Techniques and Outcomes

Gross total resection (GTR) remains the cornerstone of TIG management, but the tumor’s proximity to eloquent areas and neurovascular structures presents significant challenges. Biswas et al. (2024) [13] reported that GTR was achieved in only 39% of cases, with factors such as Zone II involvement and lack of contrast enhancement associated with lower resection success rates [13,24].

Das et al. (2022) [18] highlighted the importance of utilizing anatomical landmarks and advanced microsurgical techniques to minimize complications like ischemia and neurological deficits [18]. Awake craniotomy, as described by Tan et al. (2024), allows real-time monitoring of speech, motor, and sensory functions during surgery, significantly reducing postoperative morbidity [16]. This technique has proven particularly effective in preserving eloquent cortical areas and vascular integrity.

Despite these advancements, achieving a balance between aggressive tumor resection and functional preservation remains a major concern. Tailored approaches leveraging advanced imaging and intraoperative functional monitoring are critical to improving outcomes while minimizing complications (Figure 4) [25].

Advances in surgical techniques, including minimally invasive methods, are emerging as promising alternatives in the treatment of temporo-insular gliomas (TIGs). These approaches aim to minimize surgical trauma and expedite recovery, although their effectiveness in TIG management is still under investigation. Intraoperative tools such as neuronavigation and real-time ultrasound have significantly enhanced surgical precision by providing detailed anatomical guidance, improving both patient safety and outcomes [25,26].

Adjuvant treatments, particularly radiation and chemotherapy, remain cornerstones in the management of high-grade gliomas, including glioblastoma (GBM). The combination of temozolomide and radiotherapy is the established standard of care, offering substantial improvements in survival rates. Studies by Tan et al. (2024) and Biswas et al. (2024) [13] highlight the enhanced efficacy of this combination, especially in patients with a methylated MGMT promoter, a biomarker predicting better responses to temozolomide [13,16].

Despite these advancements, systemic toxicity and eventual resistance to temozolomide continue to pose significant challenges in long-term disease control. This has led to growing interest in targeted therapies and immunotherapies. Mandonnet et al. (2019) reported encouraging results with therapies targeting IDH-mutated gliomas, although their application in TIGs remains to be fully explored [17]. Emerging immunotherapies, including checkpoint inhibitors and cancer vaccines, offer promising prospects for overcoming treatment resistance and improving long-term outcomes [23].

The histological diversity of TIGs significantly influences outcomes. High-grade gliomas (WHO Grades III and IV), as frequently highlighted in the reviewed studies, dominate the cohort. For instance, Biswas et al. (2024) [13] reported a predominance of Grade III gliomas (77.3%) with a smaller proportion of Grade IV tumors (22.7%), while Das et al. (2022) [18] documented both astrocytomas and glioblastomas. These higher-grade tumors often present with worse prognoses and greater surgical challenges, further compounded by their infiltrative nature [13,18].

Although functional outcomes are well-documented, survival data remain sparse. Only Das et al. (2022 [18] and Capo et al. (2020) [19] provided explicit metrics, with the former reporting a median overall survival (OS) of 20 months and the latter an impressive 5-year OS of 92% [18,19]. This wide variation underscores the importance of long-term follow-up to fully understand the efficacy of current treatment protocols (Table 2).

Postoperative complications are a critical concern in TIG management, often stemming from ischemic injuries or incomplete tumor resection. Biswas et al. (2024) [13] reported persistent deficits in 10.9% of patients, primarily due to ischemic injuries. Cognitive impairments, such as memory loss and communication disorders, are also common postoperative issues, as documented by Wu et al. (2023) [21]. These findings underscore the necessity of integrating advanced imaging modalities and meticulous surgical planning to minimize risks.

Functional outcomes, as demonstrated in our case series and corroborated by the literature, emphasize the importance of preserving cognitive and motor functions. Sun et al. (2024) observed improved Karnofsky performance scores three months post-surgery, highlighting the critical role of comprehensive surgical and postoperative care in achieving favorable long-term results [23]. The high variability in postoperative functional outcomes is a key theme across studies. For example, Pitskhelauri et al. (2024) [12] reported a relatively low rate of persistent deficits at 3 months (6.3%), highlighting the role of careful surgical planning and advanced microsurgical techniques. In contrast, Das et al. (2022) [18] documented a higher rate of permanent deficits (15.2%), emphasizing the risk of ischemic complications during resection. Studies like Morshed et al. (2023) [15] demonstrated that persistent deficits were largely confined to patients with high-grade gliomas (3.8%), further emphasizing the complexity of resecting tumors in eloquent areas. Advanced tools such as triple-modality motor mapping and awake craniotomy, as highlighted by Tan et al. (2024) [16] and Wu et al. (2023), have shown significant promise in improving functional preservation [21].

The evolution of surgical strategies for TIGs has been marked by significant advancements aimed at optimizing tumor resection while preserving neurological functions. Awake craniotomy, combined with intraoperative stimulation mapping, is now considered a gold standard in many specialized centers. Research by Leroy et al. (2021) and Duffau et al. (2018) has shown that this approach reduces morbidity and enhances functional outcomes [27,28].

However, the proximity of TIGs to eloquent brain regions continues to present challenges. Even with state-of-the-art techniques, the risk of postoperative deficits cannot be entirely eliminated. This underscores the need for a careful and balanced approach, with an emphasis on meticulous preoperative planning and the use of intraoperative guidance to optimize surgical outcomes [28,29].

By combining innovative strategies with advanced imaging and precision techniques, the management of TIGs continues to evolve, offering hope for improved survival and quality of life while addressing the inherent challenges of these complex tumors [30].

### 4.2. Technological Innovations in Diagnosis and Management

TIGs have been revolutionized by advancements in neuroimaging and surgical technologies. Functional MRI (fMRI) and diffusion tensor imaging (DTI) have significantly improved preoperative planning and intraoperative navigation, allowing for precise mapping of tumor boundaries and critical neural pathways [31]. Additionally, fluorescence-guided surgery using 5-aminolevulinic acid (5-ALA) has become a critical tool, enabling real-time differentiation between tumor tissue and healthy brain tissue, thereby enhancing the precision of tumor resection [32,33,34].

Effective surgical treatment of TIGs necessitates a thorough understanding of the region’s complex spatial relationships. Rigorous anatomy training, including cadaveric dissection and microsurgical simulations, provides invaluable hands-on experience, allowing surgeons to practice techniques for preserving critical structures and navigating the intricate architecture of the insula. These training methodologies, combined with advanced tools such as neuronavigation and DTI, enhance surgical precision and enable surgeons to anticipate and mitigate intraoperative risks, such as ischemic injuries or damage to eloquent brain areas [35,36].

Mastery of insular anatomy is critical for achieving gross total resection (GTR) in TIGs while preserving neurological function. Both our findings and those in the literature underscore the delicate balance required in TIG surgeries. Comprehensive anatomy education equips surgeons with the expertise necessary to navigate this balance effectively. By integrating advanced anatomy training into neurosurgical education, we can better prepare surgeons to tackle the unique challenges posed by TIGs, ultimately improving patient outcomes and quality of life [37] (Figure 5).

The integration of artificial intelligence (AI) and machine learning (ML) into neuro-oncology marks a transformative shift in the field. These technologies excel at analyzing complex datasets, offering predictive insights, optimizing surgical planning, and monitoring postoperative outcomes. Although still in its early stages of adoption, ongoing research suggests immense potential for AI and ML to significantly enhance the management of temporo-insular gliomas (TIGs) [38].

### 4.3. Patient-Specific Considerations and Personalized Outcomes

TIGs exhibit significant heterogeneity, necessitating tailored approaches to treatment. Prognosis and therapeutic strategies are influenced by factors such as tumor grade, genetic profiles, and patient-specific anatomical considerations. Molecular profiling has emerged as a cornerstone of personalized medicine, enabling targeted treatment strategies that improve patient outcomes. Advances in genomic and molecular diagnostics are increasingly guiding individualized therapeutic decisions [39,40]. Based on our institutional experience, high-grade gliomas in less eloquent regions (e.g., non-dominant frontal lobe) often allow for more aggressive resection with a lower risk of permanent deficits. Patients with a comparable tumor burden and surgical extent in these more “accessible” locations frequently experience fewer postoperative neurological impairments and achieve longer PFS/OS. Conversely, insular and perisylvian gliomas pose a higher risk of postoperative deficits due to the proximity to critical language and motor pathways. Consequently, our patients with similar WHO grades but in non-eloquent areas may have a slightly better functional outcome postoperatively and, in some instances, a modest survival advantage if GTR is more readily achieved.

### 4.4. The Importance of Interdisciplinary Collaboration and Patient Education

Effective TIG management requires collaboration among neurosurgeons, oncologists, radiologists, and other specialists. Interdisciplinary approaches ensure comprehensive and cohesive treatment plans. Moreover, patient education and involvement in shared decision-making processes foster better understanding, adherence to treatment regimens, and overall satisfaction with care [41,42,43,44].

### 4.5. Future Directions in TIG Research and Management

Multicenter Clinical Trials: Large-scale, multicenter trials are essential for evaluating the efficacy and safety of emerging therapeutic strategies for TIGs. Such studies can address the limitations of single-institution research by providing more robust and generalizable data. For instance, the roles of intraoperative MRI and 5-ALA fluorescence-guided surgery should be assessed in larger patient populations to establish standardized protocols and optimize surgical outcomes.

Molecular and Genetic Research: A deeper understanding of the molecular and genetic underpinnings of TIGs is critical for advancing the field. Research should focus on genomic profiling, identifying driver mutations, and exploring the tumor microenvironment. These efforts could lead to the development of precision medicine strategies, with therapies tailored to the unique characteristics of each tumor [29]. Another consideration is that TIGs encompass a broad spectrum of histopathological entities, ranging from lower-grade astrocytomas (WHO grade II) to glioblastoma (WHO grade IV). Although both of our case reports focused on achieving maximal safe resection, we recognize that histological grade and molecular profile (e.g., IDH mutation status, MGMT promoter methylation) substantially impact not only prognosis but also the intensity of adjuvant treatments [45]. For high-grade lesions, an aggressive surgical resection typically pairs with radiotherapy and temozolomide. Conversely, when faced with a lower-grade but eloquently situated glioma, neurosurgeons might weigh a more conservative surgical approach or adopt a staged resection strategy, aiming to preserve function while limiting tumor progression. Future studies at our institution will delve deeper into how histological subtypes influence perioperative and long-term management decisions in TIGs [46,47,48].

Long-Term Outcome Studies: Extended follow-up studies are vital to understanding the natural history of TIGs and the long-term impacts of various treatment modalities. These investigations can provide insights into survival rates, quality of life, and late-onset complications, helping to refine prognostic models and optimize long-term care protocols for survivors.

### 4.6. Technological Innovations

The exploration of cutting-edge technologies, such as the use of exoscopes in surgery and artificial intelligence (AI) in diagnostic and prognostic evaluations, represents a significant research focus. These innovations offer the potential to improve surgical precision, enhance diagnostic reliability, and enable the more accurate prediction of patient outcomes. Systematic evaluation of their integration into clinical practice will be essential to understanding their impact on surgical success rates and the quality of life for patients with TIG [34,35,36,37,38].

### 4.7. Psychosocial and Quality of Life Assessments

The psychosocial dimensions of TIGs and their effects on patients’ quality of life remain underexplored. Future research should investigate cognitive, emotional, and social challenges experienced by patients. These insights will guide the development of holistic care strategies that address not only physical health but also the psychological and social well-being of individuals living with TIG.

### 4.8. Comparative Effectiveness Research

A comparative analysis of different treatment strategies—including various surgical techniques, adjuvant therapies, and novel approaches like immunotherapy and targeted therapies—is crucial. This research will help establish evidence-based guidelines for managing TIGs, enabling optimized patient outcomes and efficient resource allocation.

### 4.9. Limitations

#### 4.9.1. Sample Size and Generalizability

The case report and literature review focus on individual or small-scale cases, which may not fully represent the broader TIG patient population. As such, conclusions drawn may have limited applicability to all TIG patients. In both of our case reports, total resection was achieved, with preservation of the gross surrounding brain tissue. However, while the first patient recovered without new deficits, the second patient experienced postoperative aphasia and right-sided weakness. This discrepancy may be attributed to several factors, including the precise location of the tumor within or near eloquent areas, the degree of histopathological aggressiveness (i.e., infiltration of subcortical tracts), and the absence of advanced functional imaging techniques (e.g., fMRI, DTI) [49]. Although conventional neuronavigation was employed, the additional use of fMRI or DTI—or an awake craniotomy with direct cortical and subcortical stimulation—could have more accurately delineated the language cortex and motor pathways, possibly reducing the risk of postoperative morbidity. In future cases, we aim to incorporate these imaging modalities into our standard surgical planning, especially for lesions adjacent to eloquent regions [50,51].

#### 4.9.2. Selection Bias

The reliance on existing literature introduces potential selection bias, influenced by inclusion and exclusion criteria, as well as publication bias, where studies with positive outcomes are more likely to be published. These factors may shape the findings and interpretations in the article.

#### 4.9.3. Methodological Diversity

Variations in study design, patient selection, treatment modalities, and outcome measures across the reviewed studies can complicate data synthesis. This methodological heterogeneity may affect the consistency of the conclusions.

#### 4.9.4. Temporal Bias

This review encompasses studies published up to a specific time point, potentially excluding the most recent advancements in TIG treatment and technology. This limitation may impact the article’s comprehensiveness and relevance.

#### 4.9.5. Regional Variations in Practice

Reported practices and outcomes may reflect regional disparities in healthcare access, technology availability, and treatment protocols, limiting the applicability of the findings to diverse geographic and healthcare settings.

#### 4.9.6. Interpretation of Complex Data

The complexity of TIGs, with their varied clinical presentations, treatment responses, and outcomes, poses challenges for data interpretation. The conclusions may be influenced by the approach taken to synthesize and analyze this multifaceted information.

#### 4.9.7. Depth of Molecular and Genetic Analysis

The discussion on molecular and genetic aspects of TIGs might not fully capture the breadth and depth of this field, as comprehensive profiling requires highly specialized studies that may not be extensively reviewed in the article.

#### 4.9.8. Follow-Up Duration

Short follow-up periods in many reviewed studies may limit insights into the long-term effects of new treatments and surgical techniques. This limitation constrains the understanding of TIG’s natural progression and the sustained efficacy of various interventions.

## 5. Conclusions

This article provides an in-depth exploration of the multifaceted challenges associated with temporo-insular gliomas (TIGs), underscoring the complexities inherent in its diagnosis, management, and treatment. The presented case report and literature review emphasize the rarity and intricacy of TIGs, advocating for a multidisciplinary approach to optimize patient outcomes.

The evolution of surgical techniques, particularly the implementation of awake craniotomy and intraoperative mapping, has marked significant progress. These innovations enhance the ability to achieve maximal tumor resection while preserving essential neurological functions. However, the risk of surgical morbidity, driven by the tumor’s proximity to eloquent brain regions, remains a critical challenge. Advancements in adjuvant therapies, including the combination of temozolomide and radiotherapy, have demonstrated substantial efficacy in managing high-grade gliomas like TIGs. Nonetheless, the persistent issues of treatment resistance and systemic side effects highlight the need for novel therapeutic approaches. Emerging strategies, such as immunotherapy and targeted therapies, hold promise but remain in exploratory phases for TIGs.

The integration of advanced neuroimaging techniques and surgical innovations, such as 5-ALA fluorescence guidance and the use of the exoscope, has been instrumental in improving surgical precision and understanding the tumor’s anatomical and functional implications. Additionally, the potential application of artificial intelligence and machine learning in TIG management offers a transformative opportunity.

## Figures and Tables

**Figure 1 curroncol-32-00041-f001:**
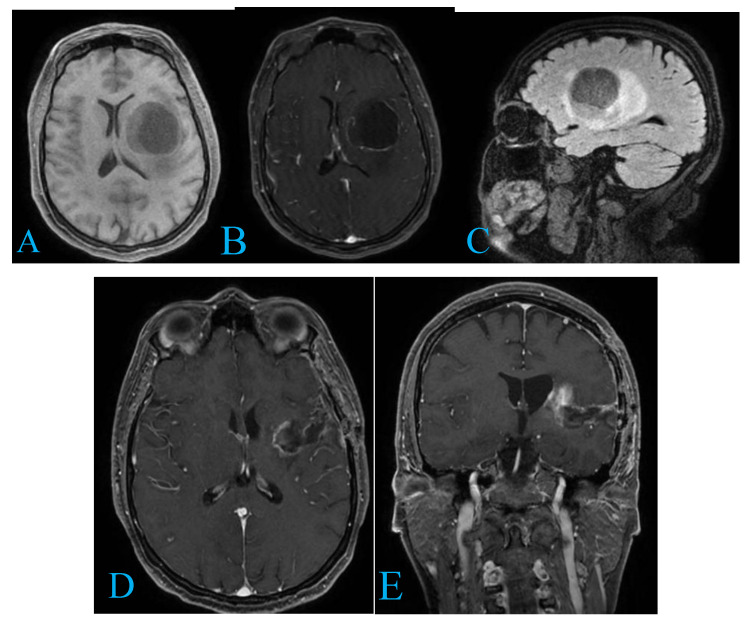
(**A**,**B**) (Axial views): Preoperative MRI scans depict a sizable temporo-insular lesion. The lesion exhibits hypointensity on T1-weighted sequences (**A**) and shows a ring-enhancing pattern on contrast-weighted sequences (**B**). A pronounced mass effect with a midline shift is also evident. (**C**) (Sagittal view): The lesion’s extent into the insular and temporal regions is clearly visualized, providing insight into its spatial involvement and proximity to critical structures. (**D**,**E**) (Postoperative axial and coronal views): Post-surgical imaging confirms gross total resection of the lesion. The images demonstrate resolution of the mass effect and midline shift, with preserved integrity of the surrounding brain tissue, emphasizing the surgical precision achieved.

**Figure 2 curroncol-32-00041-f002:**
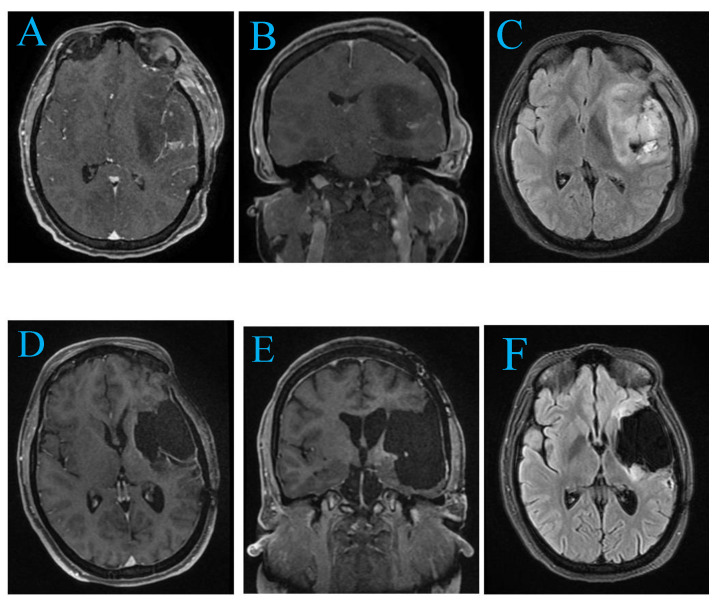
(**A**) (Axial view): Preoperative MRI depicts a heterogeneously enhancing mass in the left temporo-insular region. The mass appears hypo-intense on T1-weighted sequences, with displacement of adjacent brain structures. (**B**) (Coronal view): The mass significantly involves the insular cortex and superior temporal gyrus, exerting a mild mass effect. (**C**) (Axial view): Further delineation reveals irregular tumor borders and peritumoral edema, with compression of the left lateral ventricle. (**D**–**F**) (Postoperative axial, coronal, and axial views): Post-surgical imaging demonstrates successful tumor resection. The images show resolution of the midline shift and a substantial reduction in mass effect, with no evidence of residual tumor. Preservation of surrounding brain structures is also evident.

**Figure 3 curroncol-32-00041-f003:**
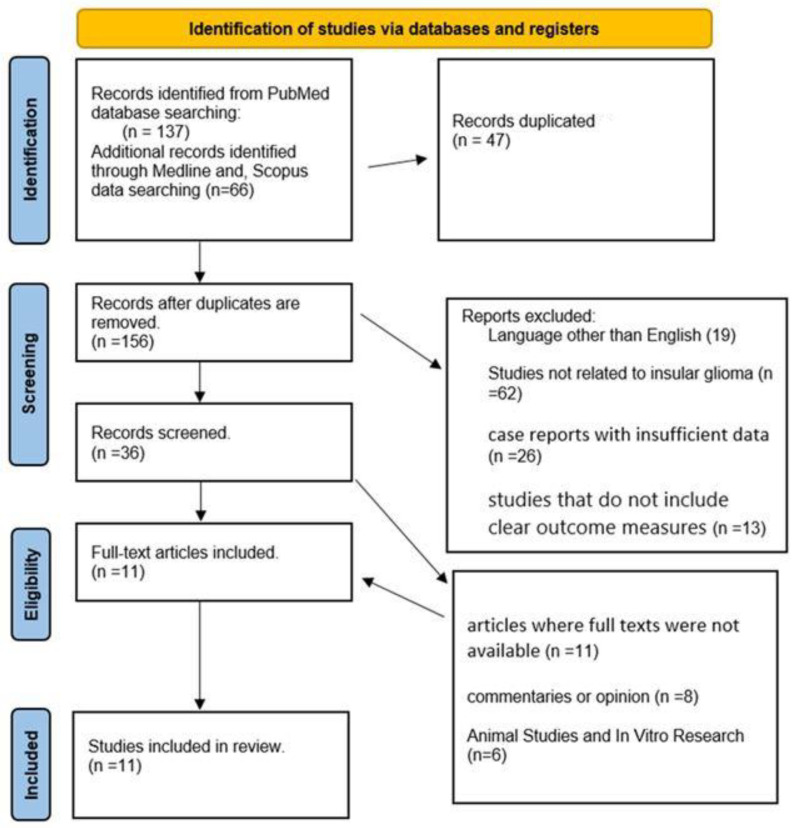
Results of the Preferred Reporting Items for Systematic reviews and Meta-Analyses (PRISMA) flow diagram illustrating the study selection process for the literature review on temporo-insular gliomas.

**Figure 4 curroncol-32-00041-f004:**
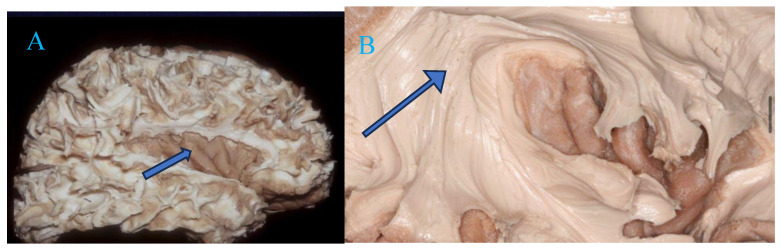
Dissection of white fibers where the insular lobe (Blue arrow) can be seen (**A**). (**B**) The Arcuate Fasciculus (Blue arrow) is the part of the Superior Longitudinal Fasciculus that extends around the insula to connect the areas of language in the inferior frontal gyrus (Broca) and the superior temporal gyrus (Wernicke).

**Figure 5 curroncol-32-00041-f005:**
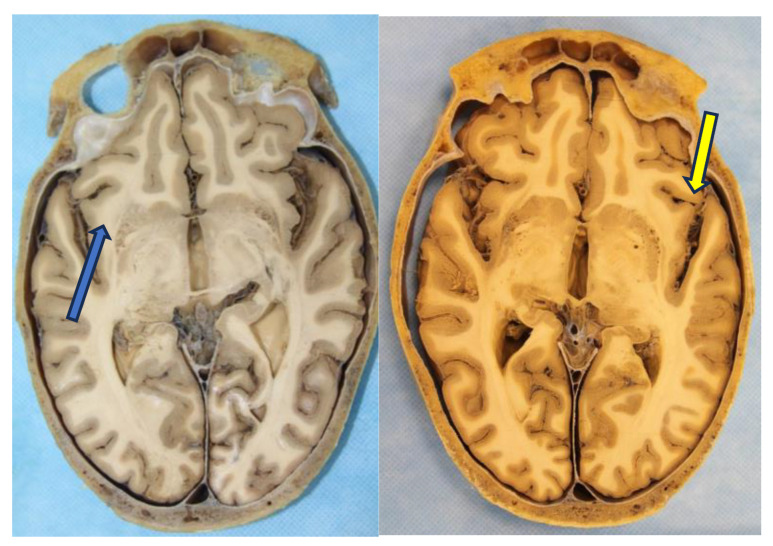
Axial section showing the relationship of the insular lobe (Blue arrow), temporal lobe, and vascular elements within Silvio’s valley (Yellow arrow).

**Table 1 curroncol-32-00041-t001:** Summary of studies reviewed on temporo-insular gliomas: diagnostic approaches, treatment modalities, complications, and outcomes.

Study Reference	Type of Study	Diagnostic Methods	Glioma Grading	Key Findings	Treatment Modalities	Complications	Patient Prognosis
Pitskhelauri, et al. (2024) [12]	Case report	MRI, histology	High grade	The most challenging part of the operation was to identify and protect the lenticulostriate arteries. Advanced microsurgical techniques and the correct patient selection for surgical treatment are cornerstones for a successful outcome.	Surgical resection, adjunct therapies.	Not applicable.	Improved after surgery and concomitant treatment.
Biswas et al. (2024) [13]	Retrospective study	Clinical data, imaging MRI, histology	High grade	Examined the clinical risk and survival in patients with glioblastoma.	Standard GBM treatment (surgery, chemotherapy, radiotherapy). Radical resection was possible in 39% of patients. Involvement of zone II and the absence of contrast enhancement predicted lower resection rate.	Persistent deficit rate was 10.9%. Overall, 45% of patients developed a postoperative infarct, 53% of whom developed deficits. Most affected vascular territory was lenticulostriate (39%).	Radical resection of insular gliomas is possible in many cases, although gross total resection is much more difficult, especially for giant lower-grade infltrative tumors involving Zone II. Ischemic injury is a major cause of defcits.
Pepper et al. (2021) [14]	Retrospective study	MRI, histology	High grade	Report on the seizure outcome after excision of insular high-grade gliomas.	Standard GBM treatment (surgery, chemotherapy, radiotherapy)	6 had poor seizure control (Engel class III/IV), and 7 died.	Median follow-up of 21 months. At long-term follow-up, of 38 patients, 23 were seizure-free (Engel class I), 2 had improved seizures (Engel class II), 6 had poor seizure control (Engel class III/IV), and 7 had died.
Morshed et al. (2023) [15]	Prospective	MRI, histology	Grades II, IIIs and IV	The authors review their results with triple-modality asleep motor mapping with motor evoked potentials and bipolar and monopolar stimulation for cortical and subcortical mapping during glioma surgery in an expanded cohort.	Surgery and chemotherapy.	Peri-resection cavity ischemia (OR 7.5, *p* = 0.04). Most persistent deficits were attributable to ischemic injury despite structural preservation of the descending motor tracts.	All persistent issues were seen in patients with high-grade gliomas, precluding detection of a statistical difference on Pearson chi-square testing.
Tan H et al. (2024) [16]	Retrospective study	MRI, histology, molecular markers	Grade IV	Awake craniotomy for glioma resection GBM/awake craniotomy for glioma resection.	Surgery.	Major complication avoidance includes recognition and preservation of eloquent cortex for language and respecting the lateral lenticulostriate arteries.	6 months postoperative were Obtained from neurooncology clinic notes.
Mandonnet E. (2019) [17]	Retrospective study	MRI, histology	IDH-mutated GBM	Report an initial experience of isocitrate dehydrogenase (IDH)-mutated insular glioma resection.	Immunotherapy, surgical resection.	None of the patients had permanent speech or motor difficulties.	Report of an initial experience of isocitrate dehydrogenase (IDH)-mutated insular glioma resection.
Das KK et al. (2022) [18]	Retrospective study	MRI, histology	Grades III, IV	Highlights the importance of anatomical landmarks in insular glioma resection and avoidance of vascular complications. We also propose to objectify the onco-functional balance in insular glioma surgery.	Surgery.	Out of seven (15.2%) patients who developed permanent neurological deficits, three (6.5%) patients had severe disability.	The median overall survival (OS) was 20 months (95% CI = 9.56–30.43). CPOI was optimal in 38 patients (82.6%).
Capo G et al. (2020) [19]	Retrospective study	MRI	High grade	Assess the neuropsychological performance in patients who undergo a recurrent surgery. In particular, we measured whether the impact of a second surgery on neuropsychological performance was significantly higher than the impact a first surgery might have.	Surgery.	No complications.	None.
Boetto J. et al. (2021) [20]	Prospective study	MRI, histopathological diagnosis	Low-grade glioma	Determine objective criteria to advocate surgical resection of an incidentally discovered suspected LGG based upon MRI findings.	Surgery.	No complications.	After 24 months no complications were described. Approximately 18.8% of incidental findings were stable over time. Insular topography, adjacent sulcal effacement, and volume greater than 4.5 cm^3^ were predictive.
Wu Z et al. (2023) [21]	Prospective study	MRI, histopathological diagnosis	Glioma type II, III and IV	Reported initial practice of using Quicktome as a pre-surgical tool to optimize the surgical sapproach in patients with insulo-Sylvian gliomas for preserving as much cognitive function as possible after surgery.	Surgical resection ultrasound for real-time guidance.	One patient experienced transient mutism postoperatively, resolving within two days. Temporary deficits were attributed to the unavoidable involvement of critical networks.	The survivorship phase is often marked by disruptive cognitive functional impairments; 30–50% of survivors encounter significant cognitive and functional impairment after treatment, which is one of the most concerning outcomes for patients.
Xue B et al. (2024) [22]	Retrospective	MRI, used Mricron (https://www.mricro.com/ accessed on 6 October 2024) to perform manual tumor mapping and calculate tumor volume.	Types II, III	Clinical features and survival outcomes of insular glioma patients are associated with our classification based on the tumor spread.	Surgery, targeted therapy.	7.8% (19/243) of the patients suffered from motor impairment, and 78.9% (15/19) patients made a functional recovery. The patients developed both motor and language disorders.	Long-term (>6 months) follow-up data of complications were available for 243 (85.9%) patients.
Sun et al. (2024) [23]	Retrospective study	MRI	Grades III, IV	The median (IQR) postoperative Karnofsky performance score 3 months after surgery was 90 (80–90). Mean temporal isthmus width was significantly higher in the affected side (involving tumor) than the contralateral one (21.6 vs. 11.3 mm; 95% CI: 9.3 to 11.3, *p* < 0.01).	Surgery.	No complications	Well tolerated in GTR, muscle strength was grade 4 or higher, and speech was nearly normal in all patients 3 months after surgery.

**Table 2 curroncol-32-00041-t002:** Summary of studies on temporo-insular gliomas (TIGs): histological grade, functional outcome, and overall survival.

Study (Reference)	Histological Grade	Functional Outcome	Overall Survival	
Pitskhelauri et al. (2024) [12]	High-grade (III and IV).	Early postoperative deficits: 31 out of 79 (39.2%) patients. Persistent deficits at 3 months: 5 out of 79 (6.3%) patients. Hence, most newly acquired postoperative deficits resolved by the 3-month mark, leaving a 6.3% persistent-deficit rate.	The study does not provide any overall survival (OS) data, progression-free survival (PFS) data, or perioperative mortality rates. The study instead focuses on (1) extent of resection (EOR), (2) early vs. persistent neurological morbidity, and (3) a new classification system correlating tumor location with EOR and postoperative outcome.	
Biswas et al. (2024) [13]	High-grade gliomas (WHO Grade 3): 77.3% of patients. High-grade gliomas (WHO Grade 4/GBM): 22.7% of patients. The majority were astrocytic tumors (IDH mutant astrocytomas). All oligodendrogliomas in this series (1p19q co-deleted) were histologically Grade 3.	The study tracked neurological outcomes at multiple time points: Immediate deficits: Defined as new or worsened deficits noted right after surgery (within the first 24 h). Delayed deficits: Deficits that developed between 24 and 48 h postoperatively. Transient deficits: Those that resolved by discharge. Prolonged deficits: Deficits persisting at 3–6 months but eventually improving. Persistent deficits: Deficits still present at or beyond 6 months (considered permanent in the context of this study).	The study reports one perioperative death (1.5% mortality, 1/66), attributable to a major vascular complication (irreversible loss of motor evoked potentials intraoperatively and postoperative MCA territory infarct).	
Pepper et al. (2021) [14]	WHO grade III (anaplastic astrocytoma, anaplastic oligodendroglioma) or WHO grade IV (glioblastoma, GBM).	Median follow-up: 21 months (mean follow-up ≈ 17–21 months). Long-term Seizure Outcomes (Engel classification): Engel I (seizure-free): 23/38 (≈60.5%). Engel II (significant improvement): 2/38 (≈5.3%). Engel III/IV (poor seizure control): 6/38 (≈15.8%). Deaths: 7/38 (≈18.4%) (these patients did not reach long-term seizure outcome assessments). Overall, 76% (Engel I or II) achieved “freedom from disabling seizures” or substantially improved seizure control.	7 died within the follow-up period (≤21 months); survival details not fully broken down.	
Morshed et al. (2023) [15]	High-grade gliomas (HGGs): 135 cases (84.4%). Low-grade gliomas (LGGs): 25 cases (15.6%).	New or worsened motor deficits by hospital discharge: 38/160 (23.8%) By 6-month follow-up, the majority of these resolved, leaving 6/160 (3.8%) patients with persistent motor deficits. All 6 persistent deficits occurred in high-grade glioma patients (4.4% of HGGs). No new persistent deficits among low-grade glioma patients (0%). Additional Observations: Language outcome: Worsened language deficit in 1/160 (0.63%). Visual outcome: Worsened visual deficit in 4/160 (2.5%).	Not reported. The median follow-up for the entire cohort was 13.2 months. The study does not report explicit overall survival (OS) or progression-free survival (PFS) metrics.	
Tan et al. (2024) [16]	High glioma (GBM).	Median intraoperative time and extent of resection were comparable between cohorts. Median KPS at initial follow-up was similar between groups (*p* = 0.650). Thus, from a broad functional standpoint (KPS), outcomes did not significantly differ when comparing ESM alone vs. ESM + HGM. Intraoperative Seizures/Afterdischarges: The prevalence of intraoperative seizure or afterdischarge events decreased in the MM cohort (12.7%) versus the ESM-only cohort (25%), though not statistically significantly (*p* = 0.150).	Not reported.	
Mandonnet (2019) [17]	IDH-mutated GBM	Extent of Resection (EOR) and Neurological Status: Median EOR: 94% (range 80–100%). Permanent Speech or Motor Deficits: None. Postoperative Ischemia: Observed in 9/12 cases (75%) on diffusion-weighted MRI, but it did not cause persistent major deficits. Neuropsychological Outcomes: Left-sided tumors: Mild deterioration in lexical abilities and verbal memory. All tumors (regardless of side): Declines in cognitive flexibility were commonly noted. Return to work: Among 9 patients who were employed preoperatively, 8 resumed their professional activity.	The abstract does not provide explicit overall survival (OS) or progression-free survival (PFS) metrics. No formal survival data (e.g., median OS, Kaplan–Meier analysis) are reported. The emphasis is on postoperative functional and neuropsychological results rather than long-term survival endpoints.	
Das et al. (2022) [18]	Astrocytomas and oligodendrogliomas Grade III and glioblastoma (GBM).	15.2% permanent deficits, 6.5% severe disability; proposed “Onco-Functional Balance” to optimize resection. At last follow-up, 32 patients (88.9%) were ILAE Grade 1 (seizure-free), 4 patients had residual seizures of varying severity.	Median OS: 20 months (95% CI: 9.56–30.43); 16/46 patients died (15 from tumor progression) during follow-up. Favorable survival predictors: Younger age < 40 years (*p* = 0.002). Non-GBM histology (*p* = 0.006). Frontal-only extension (vs. frontal + temporal) (*p* = 0.011). Caudate head involvement (*p* = 0.04).	
Capo et al. (2020) [19]	31 anaplastic astrocytomas 2 were oligodendroglioma WHO III; 7 had progressed to WHO IV (glioblastoma).	Timeline: T1/T2: pre/post first surgery (available in 17 patients). T3/T4: pre/post second surgery (full data in 40 patients). Cognitive Domains: naming, language comprehension, verbal fluency, short-term/working memory, and praxis. Main Finding: No significant overall decline from T3 (pre-second surgery) to T4 (4 months after second surgery) in the number of patients within normal range or in their mean scores. In the 17-patient subgroup evaluated over T1–T4, only phonological fluency showed a mild decline by T4. Other tasks (e.g., naming, comprehension) remained stable across both surgeries. Interpretation: Short-term (4 months) after repeated glioma surgery, major new cognitive deficits were uncommon, indicating that additional resection can be performed without significantly compromising cognitive function in the majority of patients.	The authors report a 5-year estimated overall survival (OS) of 92% in this series.	
Boetto et al. (2021) [20]	In all surgical cases, histopathology confirmed diffuse low grade.	Resection vs. watchful waiting for “incidental” suspected LGG; EOR not detailed.	No major complications reported; at 24 months, 18.8% of incidental findings remained stable without surgery.	Not reported.
Wu et al. (2023) [21]	Grades II, III, and IV.	Pre-surgical mapping with Quicktome + real-time ultrasound for resection; EOR variable.	1 patient had transient mutism, resolved within 2 days; rest had preserved cognitive function. Primary Emphasis: Evaluating cognitive (particularly “non-traditional” networks) and clinical outcomes, using the Quicktome software to map and preserve key neural pathways during surgery. Findings: By identifying and sparing these “non-traditional” or higher-order cognitive networks, the authors report better neurocognitive preservation postoperatively, though exact deficit rates or scores are not given in the citation.	Not reported.
Xue et al. (2024) [22]	Grades II and III.	Surgical resection ± targeted therapy; EOR not specifically stated.	7.8% motor deficits post-op (78.9% recovered); some had combined motor and language involvement.	The article analyzes overall survival (OS) for patients with these grade II/III insular gliomas, employing a new spread-based classification (e.g., whether the tumor crosses multiple insular zones, extends to opercula, etc.). Predictors of survival highlighted likely include tumor volume, extent of resection, IDH status, and the proposed classification (spread pattern).
Sun et al. (2024) [23]	Grades III and IV.	Transtemporal isthmus approach; reported GTR in these cases.	Post-op Karnofsky 90 (IQR 80–90) at 3 months; muscle strength ≥ grade 4; near-normal speech in all. Primary Focus: Demonstration that the transtemporal isthmus approach allows for safe exposure and resection of insular gliomas. Emphasis is on technical steps to minimize damage to eloquent structures.	Not reported.

## Data Availability

The data presented in this study are available on request from the corresponding author.

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
