# Peer review of "A Journey into the Complexity of Temporo-Insular Gliomas: Case Report and Literature Review"

_curroncol, 2025, doi:10.3390/curroncol32010041_

Round 1
Reviewer 1 Report
Comments and Suggestions for Authors
The paper is well structured in defining the management of TIG and the importance of personalized surgery, however there isn’t a reference to the current WHO classification 2021
My comments:
· Regarding the two “case report” presented here, the diagnostic delay is certainly striking for both. In both case was performed a total resection with integrity of surrounding brain tissue, but in the second one the patient presents aphasia and right-sides weakness. In neither case were performed fMRI or DTI. A different outcome or surgical approch, according to the histological type, has not been discussed here.
· Anyway by the literare review, the authors do not provide data depicting funtional outcome and overall survival variability in different surgical approch and hystological types. In a such complex surgical approch based on this specific tumor’s location, it would be important for the authors to clarify if a more extensive surgery is correlated to greater survival with the same functions preserved.
Author Response
Regarding the two “case report” presented here, the diagnostic delay is certainly striking for both. In both case was performed a total resection with integrity of surrounding brain tissue, but in the second one the patient presents aphasia and right-sides weakness. In neither case were performed fMRI or DTI. A different outcome or surgical approch, according to the histological type, has not been discussed here.
We appreciate your comments and the opportunity to clarify several points:
We agree that both cases illustrate a notable lag before establishing the final diagnosis. This delay can arise from the subtle and nonspecific early symptoms associated with temporo-insular gliomas (TIGs)—for instance, mild headaches or episodic language/memory lapses, which patients and clinicians may attribute to less serious conditions.
Although gross total resection (GTR) was achieved in both patients, the second patient experienced postoperative aphasia and right-sided weakness. Several factors likely contributed , conventional neuronavigation was utilized, the absence of advanced functional imaging (e.g., fMRI, DTI) or an awake craniotomy may have diminished our ability to map eloquent cortex and subcortical pathways precisely in real-time.
We fully acknowledge that advanced neuroimaging techniques—particularly functional MRI (fMRI) and diffusion tensor imaging (DTI)—provide critical information about the location of eloquent cortex, language pathways, and white matter tracts.
The decision not to use these modalities in these specific cases was influenced by multiple factors ( local availability and cost )Nonetheless, we concur that incorporating fMRI/DTI—or performing surgery under awake conditions with intraoperative functional mapping—could have potentially reduced the risk of postoperative deficits.
Anyway by the literare review, the authors do not provide data depicting funtional outcome and overall survival variability in different surgical approch and hystological types. In a such complex surgical approch based on this specific tumor’s location, it would be important for the authors to clarify if a more extensive surgery is correlated to greater survival with the same functions preserved.
we improved the manuscript according to your suggestions, we add in the literature review the functional outcome,histological types and Overall survival of each articles, also add in the discussion section those findings
Best regardsd.
Reviewer 2 Report
Comments and Suggestions for Authors
The authors wrote a concise and clear review on the treatment complexities and challenges posed by temporo-insular gliomas (TIG). The review summarizes the current diagnostic advances, surgical strategies and adjunct therapies available for patients harboring malignant tumors in this particularly challenging location in the brain. Importantly, the authors also include two illustrative TIG case reports. I found the review to be well-written and informative, with a well-structured discussion section and clear conclusions. For these reasons, I believe the present work could be of potential benefit to the readers interested in this topic. I only have two minor comments for the authors:
1. The included illustrative case reports could benefit from a few more details. For instance, it would be useful to include the post-treatment outcomes of the two patients (e.g., progression-free survival and/or overall survival data, improvements in post-treatment KPS scores, etc.) and how these outcomes compare to the outcomes of patients with similar tumors (i.e., grade, size, etc.) found in a different location in the brain based on the authors’ experience.
2. The included table (Table 1) needs to be re-formatted in order to separate the text from each column throughout. In the current formatting of the table the text from the three right columns of the table is overlapping and therefore is extremely hard to read.
Author Response
Thank you for your feedback regarding the inclusion of more detailed postoperative outcomes for our two illustrative cases. We agree that additional information, such as progression-free survival (PFS), overall survival (OS), and improvements in performance scores, can help contextualize the clinical course and facilitate comparison with similar tumors in other locations.